# Gender Differences in Coping and Psychological Adaptation during the COVID-19 Pandemic

**DOI:** 10.3390/ijerph20020993

**Published:** 2023-01-05

**Authors:** Rosann Cholankeril, Ellen Xiang, Hoda Badr

**Affiliations:** 1Department of Medicine, Section of Epidemiology and Population Science, Baylor College of Medicine, Houston, TX 77030, USA; 2MPH@GW, Milken Institute School of Public Health, George Washington University, Washington, DC 20052, USA

**Keywords:** COVID-19, mental health, coping strategies, anxiety, adaptive coping, maladaptive coping, psychological distress

## Abstract

This population-based study investigated gender differences in the use of coping strategies and their relationship to anxiety symptoms during the initial COVID-19 lockdown period in the United States. A national online survey was administered between 13 April 2020 and 8 June 2020. The study sample comprised 1673 respondents (66% women). Overall, 46% reported high levels of anxiety, and women experienced significantly (*p* < 0.001) higher levels of anxiety than men. Women were significantly (*p* < 0.05) more likely to use acceptance, self-distraction, positive reframing, and emotional support than men. Significant interactions between gender and coping strategies were also identified. Women engaging in high (+1SD) vs. low (−1SD) levels of active coping were not found to have significantly different anxiety levels. In contrast, men reported higher levels of anxiety when they engaged in high levels of active coping and lower levels of anxiety when they engaged in low levels of active coping (b2 = 0.88, t = 3.33, *p* = 0.001). Additionally, women engaging in high levels of acceptance and positive reframing reported significantly lower anxiety levels than when engag-ing in low levels of acceptance (b1 = −1.03, t = −4.58, *p* < 0.001) and positive reframing (b1 = −0.72, t = −3.95, *p* < 0.001). No significant associations between acceptance and positive reframing levels and anxiety levels were found with men. Overall, these findings extend our understanding of the nature of gender differences in stress responsivity during periods of high psychological distress and can inform the development of mental health interventions to respond to the COVID-19 pandemic and future infectious disease outbreaks.

## 1. Introduction

The Coronavirus 2019 (COVID-19) pandemic has been one of the defining public health crises of the 21st century. Before vaccines were developed, governments across the globe instituted social distancing, travel bans, and stay-at-home orders to mitigate viral transmission and community spread [1]. The health and safety concerns during the pandemic, combined with worldwide economic disruption and a profound alteration in social routines created a “perfect” storm for triggering psychological distress [2,3,4]. Indeed, since the start of the pandemic, there has been a dramatic rise in the prevalence and severity of mental health disorders [5,6,7,8]. In 2020 alone, there was a 27.6% increase in cases of major depressive disorders and a 25.6% increase in cases of anxiety disorders worldwide [6,9], leading some to posit that the psychological consequences of the pandemic may evolve into its most enduring health footprint [6,10]. However, research has consistently shown that the COVID-19 pandemic has affected the mental health and well-being of men and women differently, with women experiencing higher levels of anxiety [6,11,12,13,14]. A multicentric study by Fiorillo et al. [7] evaluating the impact of lockdown on mental health noted that women were significantly more likely to develop anxiety and depression-related symptoms, which would suggest significant gender differences in mental health effects. Another cross-sectional study conducted in Spain in 2020 found that women presented with greater severity of anxiety and acute stress than males [14].

Epidemiological research has shown that women experience a higher prevalence of anxiety than men. In fact, the female-to-male ratio is 2:1 for anxiety disorders [15,16]. Women are also more vulnerable to developing anxiety symptoms after traumatic or stressful events [17]. The COVID-19 pandemic may have exacerbated this pre-existing vulnerability by piling additional stressors onto women such as caregiving and household responsibilities during the initial lockdown period [18]. Gender differences in coping may also play a role [19]. Essential for mental health maintenance, Lazarus and Folkman [20] define coping as modifiable behavioral and cognitive tactics that are used to manage stress and situational demands that are appraised by an individual as distressing or taxing. Problem-focused strategies (e.g., active coping, planning, use of instrumental support) are generally viewed as adaptive due to their positive association with mental health. In contrast, emotion-focused strategies such as behavioral disengagement, denial, self-blame, and substance use are generally viewed as maladaptive due to their negative association with mental health [21,22,23,24]. Despite this, research has shown that emotion-focused coping strategies, such as acceptance, positive reframing, and seeking emotional support, are effective when dealing with stressful life circumstances such as caregiving and may even be more effective than problem-focused approaches [25,26]. Others have debated the utility of emotion-focused strategies such as denial [27] and religion [28,29]. Thus, whether a coping strategy is adaptive or maladaptive may largely depend on the contextual circumstances that prompted the need for coping in the first place as well as the observed impact of the strategy on one’s mental health.

Although the effects and consequences of the COVID-19 pandemic on mental health are fairly well documented, there is limited research on the significance of coping strategies [30]. Research conducted during the initial lockdown period found that American adults commonly dealt with stress through distraction, active coping, and seeking emotional support [31]. Worldwide, the use of some problem- and emotion-focused strategies such as positive-reframing, acceptance, and humor has been associated with better mental health during the pandemic, whereas the use of passive and avoidant emotion-focused strategies (e.g., self-blame, venting, behavioral disengagement, self-distraction) has been associated with poorer mental health [32,33,34,35]. These findings may explain gender differences in the psychological effects of the pandemic because women tend to use more passive and avoidant emotion-focused coping strategies in response to stressful situations and men tend to use more problem-focused strategies [36,37]. However, research also suggests that women who use more emotion-focused coping strategies in response to stressors report more anxiety-related symptoms compared with women who use these methods less often [38]. Likewise, women who respond to stress with avoidance or rumination tend to have higher levels of anxiety symptoms compared with men who have the same coping responses [39,40]. Women who use emotion-focused coping strategies more frequently may thus be at particular risk for higher levels of anxiety compared with men who endorse similar levels of emotion-focused coping and women who utilize these coping strategies less frequently. Such gender differences in handling stressful situations could also constitute a vulnerability that put women at increased risk for developing more anxiety symptoms during the pandemic. Studies examining mental health during the pandemic have yet to address the idea of moderated relationships between gender and choice of coping strategies. Doing so may provide valuable information on the nature of gender differences in stress responsivity and the occurrence of anxiety symptoms that could be used to guide the development of mental health interventions to respond to the COVID-19 pandemic and future infectious disease outbreaks.

Given the aforementioned issues, this study sought to examine gender differences in the use of coping styles and their relation to the presence of anxiety symptoms in a population-based sample of U.S. adults during the initial COVID-19 lockdown period. Because women are at greater risk of anxiety than men, it was expected that women would report more anxiety than men and that they would report using more emotion-focused strategies than men. It was also hypothesized that gender would moderate the relationship between the use of coping strategies and anxiety symptoms.

## 2. Materials and Methods

This study was approved by the Baylor College of Medicine Institutional Review Board (H-47505) and reports on data obtained from a study of the psychological and health behavioral impacts of the COVID-19 pandemic [41]. Eligible individuals were aged 18 years or older, resided in the U.S., and were fluent in either English or Spanish. Surveys were distributed via paid and unpaid social media advertisements and an online survey crowdsourcing platform, between 13 April 2020 and 8 June 2020. The recruitment window corresponded to the initial lockdown period that was observed by most of the U.S. [42].

A waiver of written informed consent was obtained under Department of Health and Human Services (DHHS) regulations at 45 CFR 46.117 (c) and the Common Rule. Recruitment advertisements and social media posts contained a web hyperlink that directed participants to the survey landing page, which contained a brief cover letter describing the purpose of the research, eligibility criteria, and a plain language statement. If, after reading the cover letter, individuals were interested in participating, they checked a box to confirm understanding and consent. The survey was administered on the Qualtrics survey platform (Provo, UT, USA) [43].

**Sociodemographic and Mental Health History.** Individuals were asked about their age, race/ethnicity, marital status, household income, and whether they lived with someone over 65 (yes/no) or under 18 (yes/no). They were additionally asked about their postal zip code and nearest cross streets to classify them into one of the four major US Census regions (Northeast, Midwest, South, and West). To ascertain their mental health history, individuals were asked whether they had ever received a psychiatric diagnosis (yes/no), and, if so, to specify the diagnosis.

**Anxiety.** The 4-item Patient-Reported Outcome Measure Information System (PROMIS) short form anxiety measure was administered [44,45], which assesses fear, anxious misery (e.g., worry), and hyperarousal over the past 7 days. Response options range from 1 (never) to 5 (always) and are summed to compute a raw score that can then be scaled into a T-score (standardized) with a mean of 50 and a standard deviation of 10. Scores >60 are suggestive of “caseness” and indicate the need for further psychological evaluation.

**Coping Strategies.** The Brief COPE [46] is a 28-item self-report measure that assesses 14 different problem- and emotion-focused coping strategies. Item responses range from 1 (“I don’t do this at all”) to 4 (“I do this a lot”) and scores are summed to compute scores for each subscale with higher scores indicating greater endorsement. The problem-focused strategies that were assessed in this study were: active coping, planning, and use of instrumental support. Internal consistency reliability (Cronbach’s alpha) ranged from 0.70 to 0.79, which is within what is generally considered the acceptable range of 0.65–0.80 [47,48]. The emotion-focused strategies that were assessed in this study were: acceptance, denial, positive reframing, religion, self-distraction, substance use, and use of emotional support. Internal consistency reliability was in the acceptable range (α = 0.72 to 0.92) for all of the emotion-focused subscales except self-distraction (α = 0.39). Schmitt noted that the acceptability criterion may not always be appropriate for short scales (such as the 2-item Brief Cope subscales) [49]. He further argued that when measures have other desirable properties, such as meaningful content coverage, low alphas should not deter their use. Low reliability has also been shown to affect Type II-not Type I-error, which makes it less likely to observe significant results but does not cause spuriously significant findings [50,51].

**Analysis Plan.** Descriptive statistics were calculated including means (M) and standard deviations (SD) for continuous variables, and frequencies for categorical variables. Gender differences in anxiety and use of coping strategies were examined using independent *t*-tests. A series of linear regression analyses were conducted to determine whether gender moderated associations between each of the problem- and emotion-focused coping strategies assessed by the Brief Cope and the outcome of anxiety. PROMIS anxiety scores were separately regressed on each of the coping strategies, gender, and the interaction term (coping strategy x gender) after controlling for covariates. Sociodemographic variables were included as model covariates if they were significantly (*p* < 0.05) associated with anxiety. Associations were examined using correlation analysis for continuous variables and *t*-tests and Analyses of Variance (ANOVAs) for categorical variables. Significant interactions from the linear regression analyses were probed using simple slope analysis as outlined by Preacher et al. [52]. For all analyses, effect coding was used for gender (1 = male, −1 = female), and effect sizes for significant effects were calculated using the formula *r* = [*t*^2^/(*t*^2^ + *df*)]^1/2^ [53]. All statistical analyses were performed in IBM SPSS Statistics version 28.0 (Armonk, NY, USA).

## 3. Results

### 3.1. Sample Characteristics

Of the 2435 surveys that were submitted, 213 were excluded because they did not pass our survey quality control (i.e., re-captcha, red herring questions, IP control) and data quality checks (e.g., answer consistency and speed checks). In addition, 522 (23.5%) participants of 2222, had missing data on the PROMIS anxiety measure and 25 (1.1%) did not specify male or female gender and were therefore excluded from the analyses. The resulting study sample comprised 1673 adults (1109 females and 564 males) between 18–93 years old. Table 1 depicts the sociodemographic characteristics of the study sample, stratified by gender. Overall, survey respondents were mostly female (66%), middle-aged (M = 44.67, SD = 16.2), white (63.2%), college-educated (70.9%), and resided in the Southern region of the U.S. (54.9%). A total of 137 (8.2%) respondents indicated having a pre-existing psychiatric condition, including 14 individuals with anxiety, 25 with anxiety and depression, 9 with bipolar disorder, 11 with depression, 3 with schizophrenia, 8 with posttraumatic stress disorder (PTSD), and 1 with attention-deficit/hyperactivity disorder (ADHD). Sixty-three of those who reported having a diagnosed psychiatric disorder did not specify the condition.

### 3.2. Descriptive Results

As shown in Table 2, women (t = 60.65, SD = 9.20) reported significantly (*p* < 0.001) higher levels of anxiety than men (t = 56.63, SD = 10.15), and both men’s and women’s *t*-test scores were significantly higher than the U.S. pre-pandemic population norm (M = 50.0, SD = 10.0) (*p* < 0.001). Moreover, 53.7% of women and 38.1% of men scored above the PROMIS cut-off of 60, indicating significant levels of anxiety warranting further psychological evaluation.

As Table 2 also depicts, in partial support of our hypothesis, women engaged more in the emotion-focused coping strategies of acceptance, self-distraction, and use of emotional support. However, men engaged more in substance use and denial, and there were no significant gender differences in the use of religion as a coping strategy. Of the problem-focused coping strategies, women engaged more in planning and positive reframing than men, while there were no significant gender differences in the use of active coping and the use of instrumental support.

### 3.3. Regression Analysis

A series of multiple regression analyses were conducted to examine anxiety symptoms as a function of the different problem- and emotion-focused coping strategies and gender, after controlling for age, marital status, and living with someone over the age of 65.

Problem-focused Coping. As Table 3 shows, significant interactions with gender were found for active coping, planning, and use of instrumental support.

As Figure 1a shows, women had higher levels of anxiety than men, regardless of their use of active coping, and the difference between those who were high (+1SD) vs. low (−1SD) on active coping was not significant (b_1_ = 0.004, t = 0.020, *p* = 0.984). In contrast, men reported higher levels of anxiety when they engaged in high levels (+1SD) of active coping and lower levels of anxiety when they engaged in low levels (−1SD) of active coping; tests of the simple slopes showed that this difference was significant (b_2_ = 0.88, t = 3.33, *p* = 0.001).

As Figure 1b illustrates, tests of the simple slopes showed that both men (b_2_ = 1.20, t = 4.63, *p* < 0.001) and women (b_1_ = 0.57, t = 2.96, *p* = 0.003) reported significantly higher levels of anxiety when they engaged in higher levels (+1SD) of planning as a coping strategy compared to when they used lower levels (−1SD) of planning.

As Figure 1c shows, both men and women reported higher levels of anxiety when they used more (+1SD) instrumental support compared to when they used less (−1SD) instrumental support, and tests of the simple slopes showed that this difference was significant for both women (b_1_ = 0.69, t = 3.45, *p* = 0.001) and men (b_2_ = 1.68, t = 6.72, *p* < 0.001).

Emotion-focused coping. As Table 3 shows, significant main effects were found for denial (effect size r = 0.21), self-distraction (effect size r = 0.19), and substance use (effect size r = 0.23), with greater use of these coping strategies being associated with significantly higher levels of anxiety (all *p*’s < 0.001). Significant interactions with gender were also found for acceptance, religion, and the use of emotional support.

As Figure 2a shows, when women engaged in high (+1SD) levels of acceptance, they reported lower levels of anxiety and when they engaged in low (1SD) levels of acceptance, they reported higher levels of anxiety. Tests of the simple slopes showed that this difference was significant (b_1_ = −1.03, t = −4.58, *p* < *0*.001). In contrast, men reported slightly greater anxiety when they engaged in high levels of acceptance compared to when they engaged in low levels of acceptance, but tests of the simple slopes showed that this difference was not significant (b_2_ = 0.28, t = 1.01, *p* = 0.31).

As Figure 2b shows, women reported higher levels of anxiety when they engaged in low levels (−1SD) of positive reframing and lower levels of anxiety when they engaged in high levels (+1SD) of positive reframing. Tests of the simple slopes showed that this difference was significant (b_1_ = −0.72, t = −3.95, *p* < 0.001). In contrast, men reported slightly greater anxiety when they engaged in high levels of positive reframing compared to when they engaged in low levels of positive reframing, but tests of the simple slopes showed that this difference was not significant (b_2_ = 0.21, t = 0.80, *p* = 0.43).

As Figure 2c illustrates, tests of the simple slopes showed that only men (b_1_ = 0.40, t = 1.84, *p* = 0.07) reported higher levels of anxiety when they engaged in high levels (+1SD) of religious coping compared to when they used low levels (−1SD) of religious coping.

As Figure 2d shows, both men and women reported greater anxiety when they used more (+1SD) emotional support compared to when they used less (−1SD) emotional support. The tests of the simple slopes showed that this difference was significant for men (b_1_ = 1.31, t = 5.37, *p* < 0.001) and women (b_2_ = 0.38, t = 2.05, *p* = 0.04).

## 4. Discussion

In this population-based study, which is among the first to investigate gender differences in coping strategies among U.S. adults and their association with anxiety levels during the initial lockdown period of the COVID-19 pandemic, we found that 46% of survey respondents were experiencing significant anxiety symptoms, warranting further psychological evaluation. Consistent with previous studies, women reported higher levels of anxiety than men [13,15,16]. Likewise, in partial support of our hypothesis, women were more likely to use the emotion-focused coping strategies of acceptance, self-distraction, and emotional support than men [36,37,38]. Contrary to most pre-pandemic research on the use of coping strategies in times of stress, the use of most of the problem- and emotion-focused strategies we examined was associated with greater anxiety for either men, women, or both [19,21,24,25,26]. The exceptions were acceptance and positive reframing, which were associated with lower levels of anxiety for women only. These results suggest that men and women used different coping strategies during the pandemic lockdown period and that their choice of strategies contributed to increased vulnerability (in the case of men) for developing anxiety and increased resilience (in the case of women) in the face of anxiety. The differences in the way women cope with stress could be related to the increased prevalence of anxiety symptoms and disorders relative to men [54]. Overall, these findings support our understanding of the nature of gender differences in stress responsivity and can inform the development of mental health interventions to respond to the COVID-19 pandemic and future infectious disease outbreaks.

The results of this study demonstrated that even though women experienced significantly more anxiety symptoms than men during the initial COVID-19 lockdown period, they were also more likely to engage in a variety of problem- and emotion-focused coping strategies than men. Several studies have found that women tend to use more emotion-focused coping strategies and men use more problem-focused strategies for dealing with stressful experiences [54,55,56]. Previous studies have shown that women disproportionately experienced added stressors during the pandemic, such as the pressures of working from home coupled with childcare responsibilities, that may have prompted them to venture beyond gendered coping tendencies and explore more problem-focused coping strategies to manage their stress [5,10,57,58]. What is interesting, however, is that the use of most problem-focused coping strategies was associated with increased anxiety for both men and women. This may be due to the fact that planning and soliciting practical help from others were extremely difficult to do given the pandemic and lockdown period in which people had to stay at home and were socially isolated.

In this study, the use of some coping strategies (i.e., planning, religion, instrumental support, emotional support) was associated with higher anxiety levels for both men and women, whereas the use of other coping strategies was associated with higher anxiety levels for men only (e.g., active coping). The use of acceptance and positive reframing was also found to be beneficial in terms of lower levels of anxiety for women but not for men. Similarly, a 2008 study found higher levels of depressive symptoms are associated with the use of less positive reframing in women compared with men [59].

These findings suggest that the gendered tendency for men to engage in active coping (a problem-focused strategy) may have been a risk factor for their development and experience of anxiety during the pandemic. In contrast, women’s gendered tendency to favor acceptance (an emotion-focused strategy) and positive reframing may have been protective in terms of buffering them from the adverse effects of the pandemic on their mental health. Indeed, in previous studies, the use of acceptance under circumstances involving acute stress (such as the pandemic) has been shown to be significantly related to lower levels of psychological distress [60,61]. Thus, in the case of the pandemic, women who practiced acceptance and positive reframing may have been better able to acclimate to the pandemic than those who were less accepting or more pessimistic. In a similar vein, active coping involves directly working to control a stressor through problem-solving and seeking information. Given the high degree of distress and constantly changing information landscape in the early days of the pandemic, the use of this coping strategy may have caused frustration and exacerbated men’s anxiety. In addition, research has shown that women are more likely to seek out emotional support than men [62,63], so men who went against this gendered tendency during the pandemic may have been experiencing increased levels of distress. Future research on the comparative effectiveness of acceptance versus other coping strategies in response to pandemic-related stress for men and women would provide further information for the creation of effective prevention and intervention programs that target gender differences in the presentation of anxiety symptoms and disorders.

Gender did not significantly moderate associations between some of the emotion-focused coping strategies (i.e., denial, self-distraction, substance use) that we examined, even though the *t*-tests (Table 2) revealed that the use of the strategies differed significantly between men and women. It is therefore important to keep in mind that the pandemic represents an unprecedented life stressor and our results do not rule out the presence of moderated relationships under other circumstances. An advantage of this study was the large sample size, which was racially, ethnically, and socioeconomically diverse, and its focus on gender differences in the use of coping styles and their relation to anxiety symptoms in a population-based sample of U.S. adults during the initial COVID-19 lockdown period. However, participant responses are subject to self-selectivity and sampling bias given that the nature of the study is dependent on voluntary subject participation. It is also important to note that the study’s design as a cross-sectional survey is limited in its ability to evaluate potential long-term outcomes. A longitudinal study design is needed to assess responses to coping strategies. Finally, the study’s findings are consistent with other pandemic-type situations, but we also recognize that the data was collected in a period of acute stress, which may make comparison incongruous. Research during other types of stressful events or with clinical populations may shed light on whether such effects play a role in the development of anxiety symptoms and disorders.

## 5. Conclusions

Overall, we hope our findings will inform policy discussions regarding how a mental health care system already struggling to meet the treatment need prior to this pandemic [64] may now need to accommodate increased demand. We need better systems for monitoring psychosocial needs and bolstering access to mental health services such as telehealth. More comprehensive insurance coverage for phone and video psychotherapy and allowing license reciprocity to enable telehealth across state lines could go a long way toward improving mental health care during the pandemic and beyond [65,66,67]. Likewise, public health campaigns that normalize anxiety in reaction to highly stressful events, promote self-care resources and strategies, and disseminate useful information on how to access mental health services will also be critical [66,68]. Taking such steps could increase access to those experiencing mental health concerns or who live in areas with few mental health providers.

## Figures and Tables

**Figure 1 ijerph-20-00993-f001:**
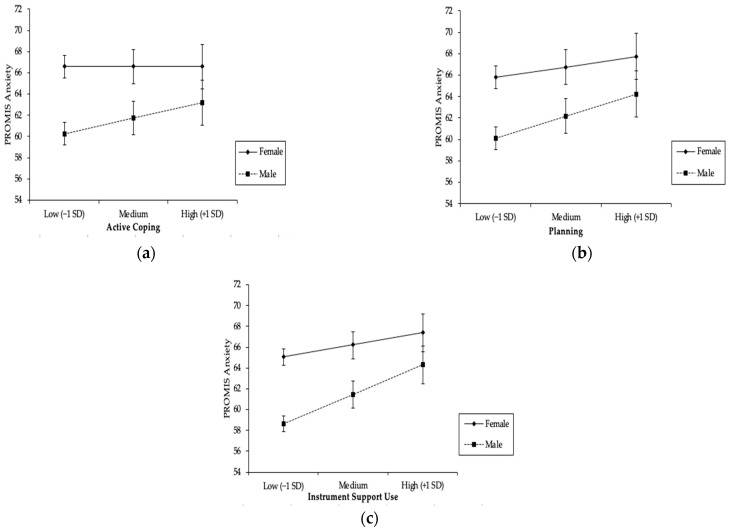
Results of Linear Regression Analysis Showing PROMIS Anxiety Scores as a Function of Problem-focused Coping Strategies and Gender: (**a**) moderation by gender on the effect of active coping on PROMIS Anxiety; (**b**) moderation by gender on the effect of planning on PROMIS Anxiety; (**c**) moderation by gender on the effect of instrument support use on PROMIS Anxiety.

**Figure 2 ijerph-20-00993-f002:**
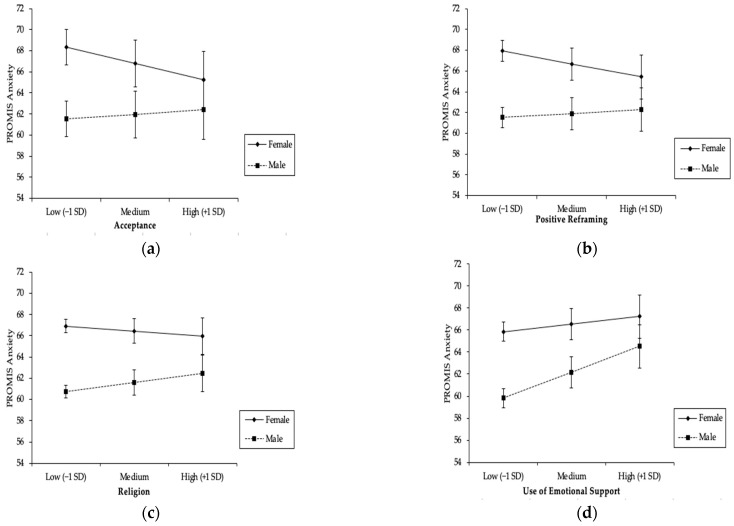
Results of Linear Regression Analysis Showing PROMIS Anxiety Scores as a Function of Emotion-focused Coping Strategies and Gender: (**a**) moderation by gender on the effect of acceptance on PROMIS Anxiety; (**b**) moderation by gender on the effect of positive reframing on PROMIS Anxiety; (**c**) moderation by gender on the effect of religion use on PROMIS Anxiety; (**d**) moderation by gender on the effect of emotional support use on PROMIS Anxiety.

**Table 1 ijerph-20-00993-t001:** Sample Characteristics (*n* = 1673).

	Gender		
	Female *n* = 1109 Number (Percent)	Male *n* = 564 Number (Percent)	Total *n* = 1673 Number (Percent)
**Race/Ethnicity**			
White	787 (70.1)	271 (48.0)	1058 (63.2)
Black	86 (7.5)	171 (30.3)	257 (15.4)
Hispanic	121 (10.9)	76 (13.5)	197 (11.8)
Other	108 (9.82)	42 (7.4)	150 (9.0)
**Education**			
Non-College Educated	269 (23.8)	214 (37.9)	483 (28.9)
College Educated	838 (75.6)	349 (61.9)	1187 (70.9)
**Marital Status**			
Unmarried	489 (44.1)	270 (47.9)	759 (45.4)
Married	620 (55.9)	293 (51.9)	913 (54.5)
**Prior History of Psychiatric Illness**			
Yes	103 (9.3)	34 (6.0)	137 (8.2)
**Annual Household Income**			
Less than $25 K	140 (12.6)	80 (14.2)	220 (13.2)
$25,000 to $74,999	349 (31.5)	195 (34.6)	544 (32.5)
$75 K or more	572 (51.6)	275 (48.8)	847 (50.6)
**Lives with someone > age 65**			
Yes	170 (15.3)	122 (21.6)	292 (17.5)
**Lives with someone < age 18**			
Yes	384 (34.6)	202 (35.8)	586 (35.0)
**U.S. Region of Residence**			
Northeast	154 (13.9)	115 (20.4)	269 (16.1)
Midwest	153 (13.8)	87 (15.4)	240 (14.3)
South	647 (58.3)	271 (48.0)	918 (54.9)
West	134 (12.1)	86 (15.2)	220 (13.2)

**Table 2 ijerph-20-00993-t002:** Descriptive Results for Women and Men.

													Women		Men		
	1	2	3	4	5	6	7	8	9	10	11	12	Mean ±SD	Range	Mean ± SD	Range	t
1. Age	--	−0.26 ***	−0.06	−0.04	−0.06	−0.13 **	0.12 **	−0.23 ***	−0.08	−0.15 ***	−0.28 ***	−0.05	44.91 ± 15.15	18.00–86.00	44.42 ± 17.24	18.00–93.00	−0.57
2. PROMIS ANXIETY	0.21 ***	--	0.19 ***	0.24 ***	0.08	0.31 ***	0.05	0.30 ***	0.12 **	0.26 ***	0.37 ***	0.24 ***	60.65 ± 9.19	40.30–81.60	56.63 ± 10.15	40.30–81.60	−7.89 ***
3. Active Coping	0.02	−0.005	--	0.58 ***	0.52 ***	0.42 ***	0.36 ***	0.23 ***	0.37 ***	0.39 ***	0.16 ***	0.38 ***	4.92 ± 1.65	2.00–8.00	4.92 ± 1.76	2.00–8.00	−0.08
4. Planning	−0.04	0.14 ***	0.59 ***	--	0.55 ***	0.45 ***	0.41 ***	0.26 ***	0.34 ***	0.32 ***	0.22 ***	0.41 ***	5.15 ± 1.71	2.00–8.00	4.93 ± 1.78	2.00–8.00	−2.26 *
5. Positive Reframing	−0.05	−0.11 ***	0.43 ***	0.42 ***	--	0.43 ***	0.37 ***	0.32 ***	0.47 ***	0.39 ***	0.25 ***	0.45 ***	5.02 ± 1.80	2.00–8.00	4.75 ± 1.78	2.00–8.00	−2.75 **
6. Use of Instument. Support	−0.15 ***	0.18 ***	0.31 ***	0.44 ***	0.32 ***	--	0.21 ***	0.48 ***	0.48 ***	0.38 ***	0.36 ***	0.68 ***	4.16 ± 1.63	2.00–8.00	4.08 ± 1.80	2.00–8.00	−0.85
7. Acceptance	−0.003	−0.12***	0.33 ***	0.36 ***	0.38 ***	0.18 ***	--	−0.09 *	0.24 ***	0.37 ***	0.03	0.31 ***	6.45 ± 1.47	2.00–8.00	6.02 ± 1.67	2.00–8.00	−5.06 ***
8. Denial	0.003	−0.23 ***	−0.02	0.01	−0.02	0.08 **	−0.23 ***	--	0.37 ***	0.17 ***	0.51 ***	0.37 ***	2.67 ± 1.24	2.00–8.00	3.30 ± 1.72	2.00–8.00	7.35 ***
9. Religion	0.13 ***	−0.09 **	0.24 ***	0.18 ***	0.32 ***	0.17 ***	0.13 ***	0.09 **	--	0.21 ***	0.20 ***	0.36 ***	4.57 ± 2.24	2.00–8.00	4.48 ± 2.11	2.00–8.00	−0.80
10. Self-distraction	−0.16 ***	0.28 ***	0.27 ***	0.29 ***	0.22 ***	0.30 ***	0.15 ***	0.13 ***	0.10 ***	--	0.28 ***	0.40 ***	5.38 ± 1.62	2.00–8.00	5.05 ± 1.65	2.00–8.00	−3.72 ***
11. Substance Use	−0.15 ***	0.23 ***	0.04	0.05	0.02	0.10 **	0.007	0.22 ***	−0.10**	0.18 ***	--	0.27 ***	2.88 ± 1.52	2.00–8.00	3.56 ± 1.91	2.00–8.00	6.92 ***
12. Use of Emotional Support	−0.11 ***	0.11 ***	0.29 ***	0.35 ***	0.29 ***	0.62 ***	0.23 ***	0.005	0.17 ***	0.34 ***	0.12 ***	--	4.81 ± 1.75	2.00–8.00	4.34 ± 1.82	2.00–8.00	−4.77 ***

Note: Women’s correlations on lower diagonal, Men’s correlations on upper diagonal; SD = standard deviation, t = *t*-test, * *p* < 0.05; ** *p* < 0.01; *** *p* < 0.001.

**Table 3 ijerph-20-00993-t003:** Effects of Coping Strategies and Gender on PROMIS Anxiety.

	β	SE	t	95% Confidence Interval		Effect Size r
				Low	High	
Problem-Focused Coping Strategies						
Active Coping	0.44	0.17	2.68 *	0.12	0.77	0.07
Gender	−4.60	0.87	−5.27 ***	−6.32	−2.89	0.13
Active Coping × Gender	0.44	0.17	2.65 *	0.11	0.77	0.06
Planning	0.88	0.16	5.47 ***	0.57	1.20	0.13
Gender	−3.92	0.86	−4.55 ***	−5.62	−2.23	0.11
Planning × Gender	0.32	0.16	1.96 *	0.00	0.63	0.05
Use of instrumental support	1.19	0.16	7.38 ***	0.87	1.50	0.18
Gender	−4.43	0.71	−6.22 ***	−5.58	−3.03	0.15
Use of Instrumental Support × Gender	0.50	0.16	3.12 **	0.19	0.81	0.08
Emotion-Focused Coping Strategies						
Acceptance	−0.37	0.18	−2.07 *	−0.72	−0.02	0.05
Gender	−6.53	1.16	−5.65 ***	−8.80	−4.27	0.14
Acceptance × Gender	0.65	0.18	3.65 ***	0.30	1.01	0.09
Denial	1.64	0.18	9.02 ***	1.28	1.99	0.21
Gender	−2.38	0.61	−3.90 ***	−3.58	−1.18	0.09
Denial × Gender	−0.15	0.18	−0.81	−0.51	0.21	0.02
Positive Reframing	−0.25	0.16	−1.60	−0.57	0.06	0.04
Gender	−4.69	0.83	−5.65 ***	−6.32	−3.06	0.14
Positive Reframing × Gender	0.46	0.16	2.92 **	0.15	0.77	0.07
Religion	0.09	0.13	0.70	−0.17	0.35	0.02
Gender	−3.82	0.66	−5.77 ***	−5.11	−2.52	0.14
Religion × Gender	0.31	0.13	2.35 *	0.05	0.57	0.06
Self-Distraction	1.39	0.17	8.10 ***	1.05	1.72	0.19
Gender	−1.49	0.92	−1.62	−3.30	0.32	0.04
Self-Distraction × Gender	−0.13	0.17	−0.75	−0.46	0.21	0.02
Substance Use	1.56	0.16	9.73 ***	1.24	1.87	0.23
Gender	−3.60	0.58	−6.18 ***	−4.75	−2.46	0.15
Substance Use × Gender	0.23	0.16	1.44	−0.08	0.53	0.04
Use of Emotional Support	0.85	0.15	5.50 ***	0.55	1.15	0.13
Gender	−4.37	0.75	−5.79 ***	−5.85	−2.89	0.14
Use of Emotional Support × Gender	0.47	0.15	3.03 **	0.16	0.77	0.07

Note: β = standardized coefficient; SE = standard error, t = *t*-test, effect size r = [t^2^/(t^2^ + df)]^1/2^, * *p* < 0.05; ** *p* < 0.01; *** *p* < 0.001.

## Data Availability

Not applicable.

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
