# Peer review of "Gender Differences in Coping and Psychological Adaptation during the COVID-19 Pandemic"

_ijerph, 2023, doi:10.3390/ijerph20020993_

Round 1

Reviewer 1 Report

Thank you for the opportunity to review this paper. I think this is a good paper, but there are several major flaws that must be addressed for this work to be of greatest value to the scientific community.

1) the conflation of biological sex, sex assigned/assumed at birth, and gender terms is the most pressing error that must be addressed especially in the hopes of describing "sex differences". If you are going to describe a sex/gender binary comparison then it would be best if you clearly define the difference you are hoping to describe and the groups that you are describing it across--use sex terms when describing sex (e.g. male and female) and gender terms when describing gender (e.g. man and woman)

2) the lack of description regarding the items collecting information about "sex" or "gender" makes this study irreproducible.

3) The introduction/background must be inclusive of the large range of information surrounding psychosocial factors that exhibited sex and gender differences during the pandemic.

4) because uncertainty is a generally well defined concept on its own, I would suggest that unless you measured it during the pandemic that you not suggest that it played a role in the impacts of the pandemic on coping and psychological adaptation. While we may agree that there was and likely remains uncertainty surrounding the SARS-CoV-2 virus unless you understand the individual's uncertainty then it is less accurate to discuss it as a major factor in the findings you report.

5. In the discussion you suggest that some associations were not moderated  (significantly), but then say that use of the strategies may not have differed between men and women--you should have evaluated differences in these as part of your preliminary bivariate analyses. If you did not then you should do that and report it. 

6. Finally the analyses fail to indicate that you controlled for individual's having a prior psychiatric diagnosis, seeing that this was data you collected and likely a contributor to how people may approach distressing events you need to address this.

Author Response

We thank the editor and reviewers for their thoughtful review of our paper. We have made the requested edits and believe that by doing so, our manuscript is much improved. Below, we provide a point-by-point response to the issues raised.

1) the conflation of biological sex, sex assigned/assumed at birth, and gender terms is the most pressing error that must be addressed especially in the hopes of describing "sex differences". If you are going to describe a sex/gender binary comparison then it would be best if you clearly define the difference you are hoping to describe and the groups that you are describing it across--use sex terms when describing sex (e.g. male and female) and gender terms when describing gender (e.g. man and woman)

Response: We thank the reviewer for pointing out this distinction, and we concur that gender terminology must be addressed. For this study, we are evaluating gender differences since, in the survey, we asked participants to self-identify their gender. We have removed any mention of biological sex and updated to the appropriate terminology in the title and throughout the manuscript.

2) the lack of description regarding the items collecting information about "sex" or "gender" makes this study irreproducible.

Response: We agree with the reviewer’s comment and have made the correction. We noted gender coding and distribution on pg. 4 (line 163-164 and line 173-175).

3) The introduction/background must be inclusive of the large range of information surrounding psychosocial factors that exhibited sex and gender differences during the pandemic.

Response: We agree with the reviewer’s comment and have made the correction. We have added verbiage on the bottom of pg. 1 (line 40-47) expanding on sex differences related to psychological factors during the pandemic.

4) because uncertainty is a generally well defined concept on its own, I would suggest that unless you measured it during the pandemic that you not suggest that it played a role in the impacts of the pandemic on coping and psychological adaptation. While we may agree that there was and likely remains uncertainty surrounding the SARS-CoV-2 virus unless you understand the individual's uncertainty then it is less accurate to discuss it as a major factor in the findings you report.

Response: We agree with the reviewer’s comment that we did not measure uncertainty and removed that term accordingly on pgs. 2,10, and 11.

  1. In the discussion, you suggest that some associations were not moderated significantly but then say that use of the strategies may not have differed between men and women--you should have evaluated differences in these as part of your preliminary bivariate analyses. If you did not then you should do that and report it. 

Response: There were significant differences noted in the t-tests shown in Table 2,  and we have corrected the verbiage in the last paragraph of the discussion on pg. 10 (line 340-343) to reflect that.

  1. Finally the analyses fail to indicate that you controlled for individual's having a prior psychiatric diagnosis, seeing that this was data you collected and likely a contributor to how people may approach distressing events you need to address this.

Response: For this point, we acknowledge the reviewer’s notes. However, we did not control for individuals having prior psychiatric diagnoses as we don’t know definitively whether having a psychiatric illness had an effect on the study outcome. Moreover, controlling for individuals would restrict the sample.

Reviewer 2 Report

This empirical paper is almost ready for publication. There are still some issues which the authors should consider in finalising this paper. 

In all, this is an empirical paper, but the discussion with the very fine list of literature is almost invisible in the paper. This is not very helpful for the reader, at least the most important issues having an effect on the research questions, the design and former results should be briefly discussed in the text, not just listing the references. 

Rows 103-4, the social media advertisements and platforms used in collecting the data are mentioned. The issue of possible self-selectivity bias in responding to the survey should be at least mentioned briefly. It could be possible that persons having faced more mental pressure or problems during the pandemic are more likely to respond than others. The results are compared with previous results from the US, but are the survey results representative for the population?   

 rows 104-5: the data is collected during the lockdown period between April 13th and June 8th in 2020. Do you, however, interpret your results to hold more broadly for pandemic type of situations? This could be made clearer.

One of the background variables is marital status. You also ask, whether the persons lives with someone over 65 or under18. From the point of view of mental support and welfare, cohabiting could be more relevant than marital status.

Rows 261-4 and figure 2C, as well as rows 265-8 and figure 2D do not seem to match with each other? Please check this.

The study design does not seem to separate the timing of various coping strategies and their outcomes or the reasons for the ways of coping. It seems rather that you handle the coping strategies as an immediate response for the pandemic situation. Probably you could mention that it is another kind of study design where the longer-term outcomes of the coping strategies become visible.       

Author Response

We thank the editor and reviewers for their thoughtful review of our paper. We have made the requested edits and believe that by doing so, our manuscript is much improved. Below, we provide a point-by-point response to the issues raised.

In all, this is an empirical paper, but the discussion with the very fine list of literature is almost invisible in the paper. This is not very helpful for the reader, at least the most important issues having an effect on the research questions, the design and former results should be briefly discussed in the text, not just listing the references.

Response: We thank the reviewer for this suggestion and agree with their assessment. We have cited and restructured the first three paragraphs of the discussion found on pg. 10 in order to better contextualize the findings.

Rows 103-4, the social media advertisements and platforms used in collecting the data are mentioned. The issue of possible self-selectivity bias in responding to the survey should be at least mentioned briefly. It could be possible that persons having faced more mental pressure or problems during the pandemic are more likely to respond than others. The results are compared with previous results from the US, but are the survey results representative for the population?  

Response: We appreciate the reviewer’s comments and acknowledge possible self-selection bias at the end of the discussion on pg. 11 (line 349-351).

 Rows 104-5: the data is collected during the lockdown period between April 13th and June 8th in 2020. Do you, however, interpret your results to hold more broadly for pandemic type of situations? This could be made clearer.

Response: We do acknowledge that the results can be applied more broadly, which we now expanded upon at the end of the discussion on pg. 11(line 353-355).

One of the background variables is marital status. You also ask, whether the persons lives with someone over 65 or under 18. From the point of view of mental support and welfare, cohabiting could be more relevant than marital status.

Response: We state on pg. 4 of the analysis plan the criteria for including covariates. Sociodemographic characteristics like marital status and living with someone over 65 were included as model covariates because they were significantly associated (p < .05) with the study outcome variable. Persons living with someone under 18 were not included because it did not have a significant association with the outcome variable.

Rows 261-4 and figure 2C, as well as rows 265-8 and figure 2D do not seem to match with each other? Please check this.

Response: Thank you for catching this. It has been corrected.

The study design does not seem to separate the timing of various coping strategies and their outcomes or the reasons for the ways of coping. It seems rather that you handle the coping strategies as an immediate response for the pandemic situation. Probably you could mention that it is another kind of study design where the longer-term outcomes of the coping strategies become visible. 

Response: We agree with the reviewer’s comments on the study’s limitations. We have added language on pg. 11 (line 351-353) to address that point, as well as acknowledge the need for a long-term study to evaluate the association between coping strategies and psychological distress.

Reviewer 3 Report

Theoretical discussion is missing both in the introduction and the Discussion section.

Conclusion sounds as suggestions/discussions more than conclusion. Rewrite the Conclusion section as it is to be.

Author Response

We thank the editor and reviewers for their thoughtful review of our paper. Below, we provide a point-by-point response to the issues raised.

Theoretical discussion is missing both in the introduction and the Discussion section.

Response: We noted Lazarus and Folkman's model of stress near the top of pg. 2. Given that this study is exploratory, we did not feel it was compulsory to expand the theoretical discussion further.

Conclusion sounds as suggestions/discussions more than conclusion. Rewrite the Conclusion section as it is to be.

Response: On this point, based on the reviewer’s input, we realize that the term conclusion may have been somewhat misleading. We opted to change the title of the section to Implications for Practice instead on pg. 11 (line 382)